

# Pathological caudal skeleton of an ichthyodectiform fish from the Upper Cretaceous Niobrara Formation of western Kansas, USA

S Christopher Bennett[1]

Department of Biological Sciences, Fort Hays State University, Hays, KS, United States of America

## ABSTRACT

A series of 12 contiguous caudal vertebrae of an ichthyodectiform fish from the Smoky Hill Chalk Member of the Niobrara Formation is described. The vertebral centra exhibit extensive overgrowth of pathological bone and there is additional pathological bone within the centra and intervertebral spaces, which together resulted in the coossification of most centra. The extent of the pathology is greatest on preural vertebrae 1-3 and decreases anteriorly, which suggests that the pathology began posteriorly and progressed anteriorly. In addition to the pathological overgrowth on bones, the specimen preserves features interpreted as calcified and/or ossified soft tissues associated with the neural and haemal canals. The pathologies are unlike previously described examples of bony pathologies in fish, and it is suggested that they resulted from combined bacterial and fungal infections. As the pathologies developed, they would have adversely impacted the fish's swimming and feeding abilities, and presumably eventually led to the fish's death.

Corresponding author
S Christopher Bennett,
cbennett@fhsu.edu

## INTRODUCTION

Although pathologies are not uncommon among extant fishes (*Schlumberger & Lucké, 1948*), only a few types of pathologies have been described from fossil fishes. Most common are hyperostoses of neural and haemal spines, fin pterygiophores, and ribs that resulted in large rounded bony masses that are commonly known as 'Tilly bones' (*Gervais, 1875*; *Konnerth, 1966*; *Tiffany, Pelham & Winter, 1980*; *Schlüter, Kohring & Mehl, 1992*; *Schlüter & Kohring, 2002*; *Capasso, 2005*; *Meunier, Gaudant & Bonelli, 2010*). Such structures are common in extant fishes (*Smith-Vaniz, Kaufman & Glowacki, 1995*; *Jawad, 2013*; *Jawad, Wallace & Dyck, 2015*; *Chanet, 2018*), and can also affect skull bones and vertebrae (*Meunier, Bearez & Francillon-Vieillot, 1999*; *Béarez, Meunier & Kacem, 2005*); however, it seems that in most cases they are normal features of old individuals (*Harland & Van Neer, 2018*). A possible notochordal chondroma was described in a Late Cretaceous pycnodontid from Lebanon (*Capasso, 2022*), and though a soft tissue tumor, its presence was indicated by the displacement of neural and haemal arches. Preservation of hard nodules in the skin of fossil fish has also been reported (*Petit, 2010*; *Petit & Khalloufi,*

*2012*). Here a pathological caudal skeleton of an ichthyodectiform fish from the Upper Cretaceous Smoky Hill Chalk Member of the Niobrara Formation of western Kansas is described, which is unlike previously described pathologies in fossil fishes and, in addition to the osseous pathologies, preserves what are interpreted as calcified and/or ossified soft tissue associated with the neural and haemal canals.

## MATERIAL AND METHODS

The specimen, YPM VP 42619, is the pathological caudal skeleton of a large fossil fish collected 21 April 1876 by B. F. Mudge and his collecting party from the Santonian to early Campanian Smoky Hill Chalk Member of the Niobrara Formation of southwestern Gove County, Kansas, USA (*Hattin, 1982*). Other specimens collected on 20 and 21 April of that year (YPM VP 2390, 40541, and 40542) were identified as being collected from the 'southwest part' or 'southwest corner' of Gove County, so that presumably was the case for this specimen as well. Note that by late 1875, O. C. Marsh of the Yale Peabody Museum had told Mudge that he would no longer pay for fish specimens other than 'snout fish', *i.e., Protosphyraena* (O. C. Marsh, 1875, correspondence in YPM archives), so the fact that the specimen was collected suggests that it was deemed unusual.

The specimen was cleaned of surrounding chalk matrix using low pressure sodium bicarbonate air abrasive (*Graham & Allington-Jones, 2018*), and was examined with stereomicroscopes and photographed. A horizontal sample of bone extending to the midline was removed from the left side of one vertebral centrum, embedded, ground, and polished using standard methodology (*Chinsamy & Raath, 1992*), and the resulting thin section was examined and photographed with microscopes.

### Description

YPM VP 42619 consists of a series of 12 contiguous vertebral centra, two of which are free (*i.e.,* not fused to adjacent centra) and ten of which are coossified, most of which have broken bases of neural and haemal spines preserved in articulation with the centra (Figs. 1 and 2), and four small neural and haemal spine fragments. The posteriormost vertebra is interpreted as the first ural, the other nine vertebrae of the coossified series are interpreted as preural vertebrae 1-9, and the two free vertebrae are preurals 10 and 11. The entire series of 12 vertebrae exhibits extensive overgrowth of pathological bone tissue. The anteriormost free vertebra is 34.4 mm wide × 27.7 mm high × 18.7 mm long, the posterior free vertebra is 35.6 × 31.0 × 18.4 mm, and the two free vertebrae have a combined length of 36.9 mm. The series of ten coossified vertebrae is 172.8 mm long, and is broken into three sections: an anterior section of three vertebrae, 59.9 mm long (~20 mm/centrum), 34.4 × 36.5 mm at the anterior end, and 34.2 × 29.6 mm at the posterior end, with little taper; a middle section of four vertebrae 69.3 mm long (~17.3 mm/centrum) and 25.7 × 23.0 mm at the posterior end, with somewhat more taper; and a posterior section of three vertebrae 43.6 mm long (~14.5 mm/centrum) and 13.4 mm × 16.2 mm at the posterior end, with significant taper and a marked upward curvature posteriorly. The combined length of all 12 vertebrae is 209.7 mm. At the breaks between the sections of the coossified centra, calcite and/or bone tissue can be seen filling the biconical space between the cotyles of the adjacent

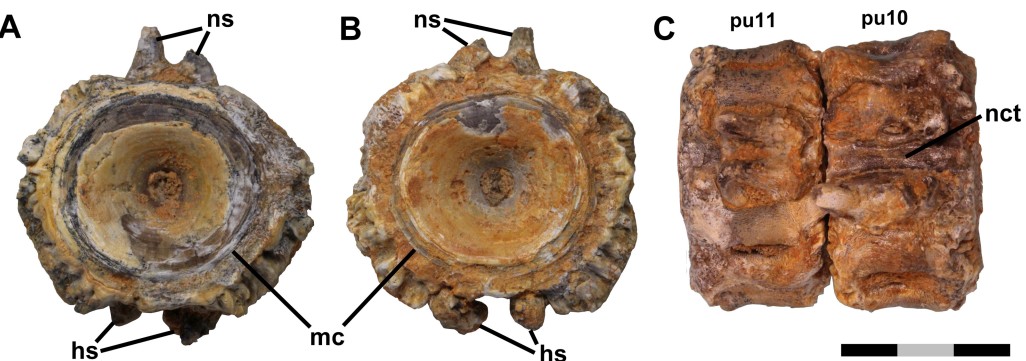

**Figure 1** **Free vertebrae.** Ichthyodectiform fish YPM VP 42619. Photographs of pathological vertebrae that were not coossified with adjacent vertebrae. Preural vertebra 11 in (A) anterior and (B) posterior views, and (C) preural vertebrae 11 and 10 articulated in dorsal view. Abbreviations: hs, haemal spine; mc, margin of cotyle; nct, calcified or ossified neural canal soft tissues; ns, neural spine; and pu, preural vertebra. Scale bar = 3 cm.

vertebrae where the intervertebral disk would have been. The posteriormost vertebra has a small broken and distorted posterior cotyle, 12.6 × 9.8 mm as preserved, which indicates that there was another vertebra posterior to it, which is why the posteriormost vertebra is interpreted as the first ural vertebra and the subsequent vertebra would have been the second ural. The fragments of neural and haemal spines not articulated with the vertebral centra consist of three 2−2.5 cm long sections of individual spines and one ∼3 cm long section of ∼5 adhering spines, which show little evidence of pathology.

The bone is mostly brown, though paler in places, presumably as a result of weathering, and parts, particularly the dorsal surfaces, are encrusted with orange ferruginous deposits. Masses of somewhat darker ferruginous particulate matter with an appearance that resembles sintered porous bronze are present within some depressions on the centra (Fig. 3A). The material seems not to be osseous, and presumably was derived from the surrounding matrix, but it resisted removal by the air abrasive when the specimen was prepared.

Anterior and posterior views of preural vertebra 11 show that the centrum is amphicoelous with an hourglass shaped cross-section typical of fishes (Fig. 1). In lateral view, the contours of the external surface of the centrum are evident despite the bony overgrowth, and include a prominent lateral longitudinal ridge bounded above and below by longitudinal grooves. That morphology is typical of ichthyodectiform fishes (*Bardack, 1965*; *Cavender, 1966*), which are represented in the Smoky Hill Chalk by *Ichthyodectes*, *Gillicus*, and *Xiphactinus*, and the morphology of the caudal skeleton of YPM VP 42619 compares well with the '*Ichthyodectes* type' caudal skeleton described and illustrated by *Cavender (1966)*, Fig. 1.

The outer margin of the anterior cotyle of preural vertebra 11 is 22.4 mm wide × 21.2 mm high, but the centrum proper is surrounded by ∼5 mm of bony overgrowth. That bone has a dense surface with an intricate reticulate pattern of ridges and grooves and many small pores opening onto the surface (Fig. 3A). Preural vertebra 10 is similar to preural 11

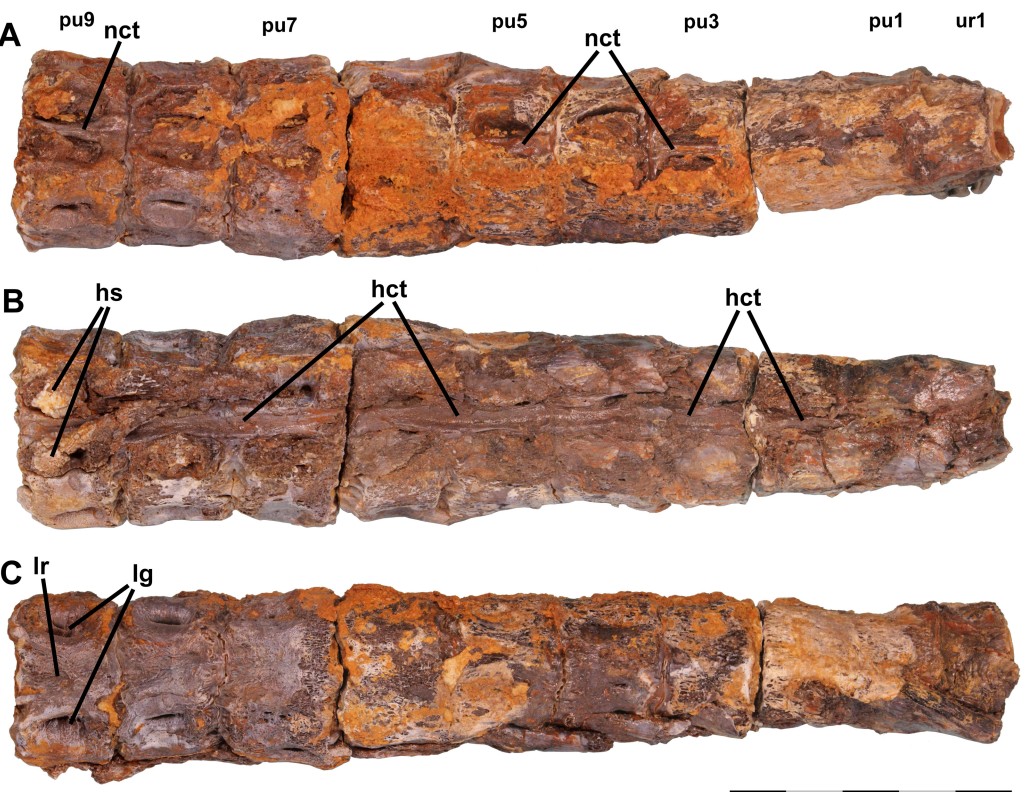

**Figure 2  Fused vertebrae.** Ichthyodectiform fish YPM VP 42619. Photographs of pathological series of coossified preural vertebrae 1-9 and ural vertebra 1 in: (A) dorsal; (B) ventral; and (C) left lateral views. Abbreviations: co, cotyle of ural vertebra 1; hct, calcified or ossified haemal canal soft tissues; hs, haemal spine; lg, lateral groove; lr, lateral ridge; nct, calcified or ossified neural canal soft tissues; pu, preural vertebra; and u, ural vertebra. Scale bar = 5 cm.

and is shown articulated with preural 11 in Fig. 1C. There is a gap between the outer parts of the overgrowing bone at the posterior end of preural 11 and the anterior end of preural 10, and the bony overgrowth on either side of the gap is thickened and coarsely textured with projections and pits. Because there was no contact between the bony overgrowth of the two vertebrae, the bony overgrowth would not have prevented movement between the two vertebrae. However, some bone tissue is present in the intervertebral space between preurals 10 and 9, which suggests that movement between those vertebrae was reduced.

The centra of the anterior section of the coossified series of vertebrae also exhibit gaps between the outermost bony overgrowth of adjacent centra (Fig. 2). However, the gaps are narrower than those between the free vertebrae, ∼2 mm or less, and at their bottoms there are sinuous grooves where the posterior surface of the bony overgrowth of the anterior vertebra meets the anterior surface of the overgrowing bone of the posterior vertebra. The grooves presumably are sinuous because they are between the bony overgrowths of the centra rather than the centra themselves and the rate of deposition of the bony overgrowth was not uniform. In some places, the bony overgrowth of adjacent vertebrae is fused and the groove obliterated, but despite the fact that in most places the bony overgrowth is

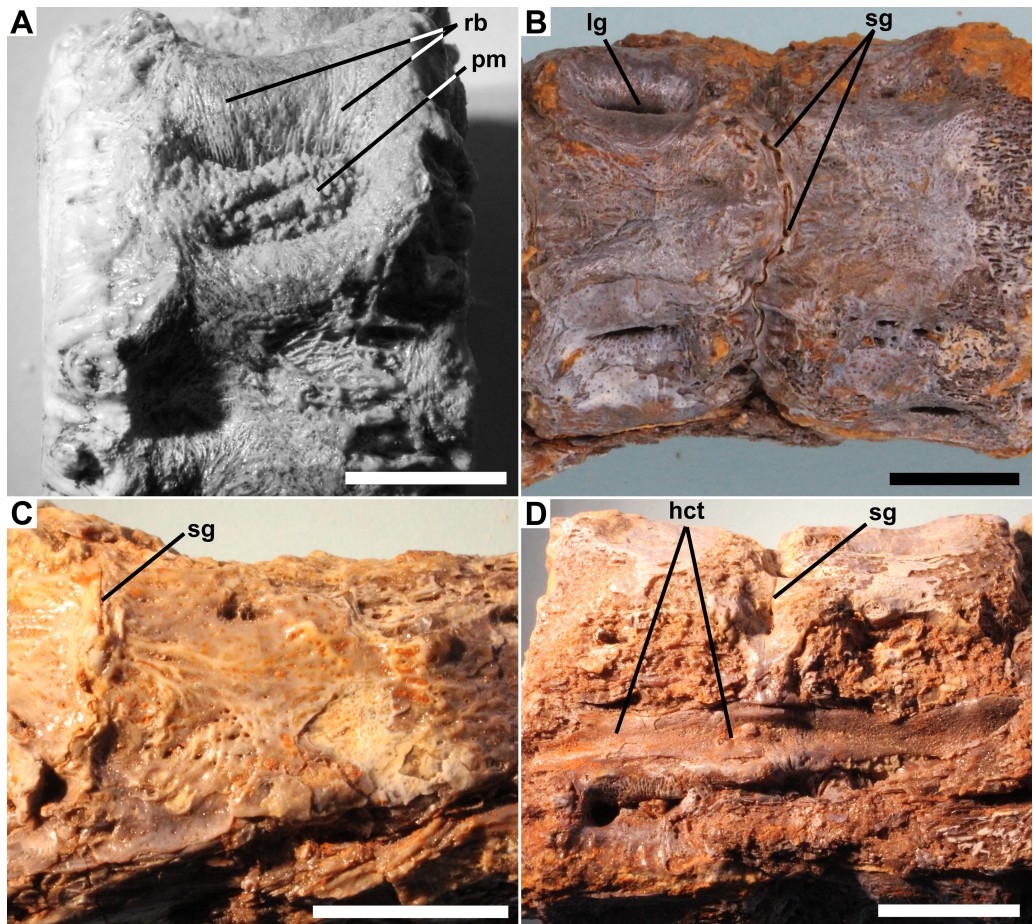

**Figure 3** **Details of pathology.** Ichthyodectiform fish YPM VP 42619. Detail photographs of selected features: (A) reticulate pattern of fine ridges and grooves on lateral surface of preural vertebra 10, and ferruginous particulate matter in longitudinal groove; (B) narrow sinuous groove between preural vertebrae 8 and 7 in ventral view (compare to Fig. 2C); (C) pathological bone tissue on right lateral surface of preural vertebra 2 through ural vertebra 1 and remnant of sinuous groove between preural vertebra 1 and ural vertebra 1; and (D) soft tissues of haemal canal on preural vertebrae 8 and 7 in ventral view with anterior to the right (compare to Fig. 2C). Abbreviations: hct, calcified or ossified haemal canal soft tissues; lg, longitudinal groove; pm, particulate matter; rb, reticulate bone; and sg, sinuous groove. Scale bars = 1 cm.

not fused, the close correspondence of the shapes of the surfaces of the adjacent vertebrae meeting at the grooves suggests that no significant movement was possible (Fig. 3C). The base of the haemal arch of preural vertebra 9 is better preserved than any other and exhibits some bony overgrowth, though to a lesser extent than the vertebral centra.

The extent of fusion of the centra increases posteriorly, and by the middle section of the coossified series of 10 vertebrae the gaps have disappeared and the narrow grooves between the bony overgrowth are at the surface of the bony overgrowth on each centrum (Fig. 3B). On the posterior section of the coossified series, the vertebrae are fully fused and the grooves obliterated. The external surface of the bony overgrowth on the posterior section appears very dense, but retains a little of the reticulate surface structure seen on the more

anterior vertebrae. Examination of the intervertebral spaces between the three sections of coossified centra found that there was bone tissue within the spaces.

Longitudinal structures are preserved on the dorsal and ventral midlines of the centra, and are presumably associated with the neural and haemal canals (Figs. 2 and 3). The ventral structure passes between the articular surfaces for, or the bases of, the haemal spines, and is best seen on the ventral midline of preural vertebrae 7–9 (Fig. 3D). The structure is ~2.5 mm wide over the middle of the centrum and ~4.0 mm wide over the intervertebral joint. At the anterior end of preural vertebra 9, the structure has a flattened suboval cross-section and does not exhibit a lumen. A similar, but somewhat smaller dorsal longitudinal structure is preserved passing between the articular surfaces for, or the bases of, the neural spines. The structures cannot be interpreted as any normally osseous part of the vertebral column proper. It seems likely that they represent calcified or ossified soft tissues associated with the haemal and neural canals, though the fact that the ventral structure widens and narrows makes it unlikely that it is the caudal vein that passes through the haemal arches in fishes (*Schultze & Arratia, 1989*). However, an alternative interpretation would be that they are pathological bone tissue that preferentially invaded and grew along the canals. If the longitudinal structures represent soft tissues, their calcification or ossification presumably was related to the osseous pathologies of the caudal skeleton, but it is not clear whether the structures were calcified or ossified, and if so, how.

## Thin section

Figure 4 shows a horizontal thin section of the left side of the centrum of preural vertebra 10 from the midline to the external surface. The conical anterior cotyle is intact and apparently was unaffected by pathology, whereas the medialmost ~25% of the posterior cotyle is missing. Between the cotyles, normal cancellous bone infilled with calcite occupies much of the centrum's volume (Fig. 4: cb), but the lateral part of the centrum is occupied by denser bone. The anterior cotyle exhibits fine striations parallel to its anterior surface that presumably are growth lines reflecting cyclical changes in the rate of deposition of bone tissue (Fig. 5A). A linear feature exhibiting similar striations that presumably were formed in a similar manner extends posteriorly at least four mm from the anterior cotyle's lateral margin and is interpreted as the former external surface of the centrum upon which pathological bone tissue was deposited. Posteriorly, there are additional features farther back that may also represent the original external surface. Based on that interpretation, the bony overgrowth surrounding the centrum ranged from 4.0 to 5.3 mm thick. The bone tissue of the overgrowth is quite dense, but exhibits various bone textures. Much of the outermost ~1 mm of bone tissue exhibits closely spaced slender subparallel canals, ~30 µm in diameter with occasional rounded expansions, and subperpendicular to the external surface such that the sectioned tissue resembles a palisade (Fig. 5B). Examination of preural vertebra 10 reveals that the palisade-like tissue is associated with the presence of the reticulate pattern on the external surface (Fig. 3A), and the small pores that are part of the reticulate pattern of that surface may be openings of the slender canals. A dark line at the base of the palisade-like layer and subparallel to the external surface may be a growth? line. Deep to the palisade-like layer, the bone tissue is generally quite dense and without any

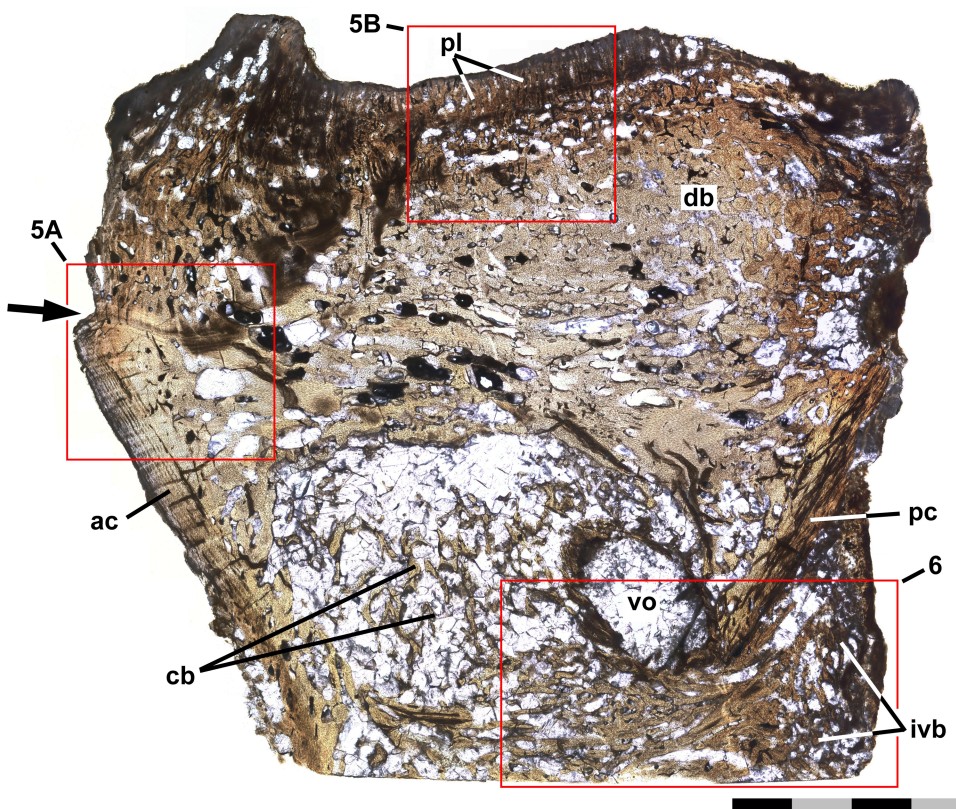

**Figure 4** **Thin section of preural 10.** Ichthyodectiform fish YPM VP 42619. Photograph of horizontal thin section of the left side of centrum of preural vertebra 10 extending laterally from the midline with anterior to the left. Numbered rectangles show the location of detail photographs in Figs. 5 and 6. The arrow at left indicates the presumed external surface of the centrum upon which pathological bone was deposited. Abbreviations: ac, anterior cotyle; ivb, intervertebral bone tissue; pc, posterior cotyle; pl, palisade-like layer; and vo, large void. Scale bar = 5 mm.

discernible pattern, but there are many spaces that are interpreted as large vascular canals. In places there are also slender subparallel canals that may represent a second palisade-like layer, and deep to them a second dark growth? line. Traces of a third layer of similar tissue with subparallel canals are visible to the left in Fig. 5B, deep to the second layer. It is not clear whether the specimen preserves successive layers of palisade-like tissue with the deeper layers being less distinct because of remodeling or the specimen records increasing expression of the palisade-like character in successive layers deposited sequentially.

The thin section also shows bone tissue in the biconical intervertebral space where the intervertebral disk would have been. The tissue is cancellous, but denser than the normal cancellous bone within the vertebral centrum that was described above. The intervertebral bone appears to be continuous with bone tissue that extends through the gap in the posterior cotyle and into the centrum near the midline (Fig. 6). The latter bone tissue exhibits a rather linear character with subparallel bands of bone and several rather large roughly longitudinal vascular canals that converge anteriorly, extend through the gap, and diverge within the centrum, where they appear to be continuous anteriorly with

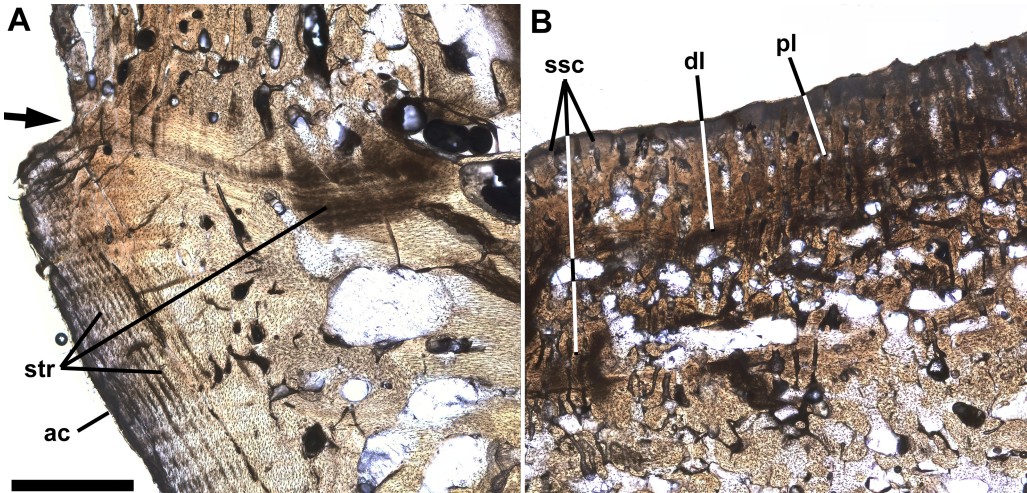

**Figure 5 Growth lines and palisade-like layer.** Ichthyodectiform fish YPM VP 42619. Detail photographs of the thin section of the centrum of preural vertebra 10 in Fig. 4 showing: (A) striations of presumably stratified bone of the anterior cotyle and similarly stratified bone of the presumed former external surface of the centrum upon which overgrowing bone tissue was deposited; and (B) thin palisade-like layer at the lateral surface with closely spaced slender subparallel canals. The arrow at left in A indicates the presumed external surface of the centrum upon which pathological bone was deposited. Abbreviations: ac, anterior cotyle; dl, dark growth? line; pl, palisade-like layer; str, striations; and ssc, slender subparallel canals. Scale bar = 1 mm.

rather dense cancellous bone tissue similar to that near the middle of the intervertebral space. It is not clear which direction the pathological bone may have grown, but it seems more likely that it was growing into the centrum from the intervertebral space, in which case, pathological bone may have been growing anteriorly through successive centra and intervertebral spaces, with what is seen in preural vertebra 10 being the leading end of what might have continued to grow into preural vertebra 11 if the fish had lived longer. One last feature of note is a large suboval void surrounded by a ring of rather dense dark appearing bone and filled with fractured calcite, which is lateral to the pathological bone passing through the gap in the posterior cotyle and just anterior to the remaining part of the cotyle (Figs. 4, 6). Presumably the void represents an abscess that resulted from an infection that destroyed all trabeculae of the cancellous bone within the ring of dense bone, but it is not clear how that process was related to the deposition of the pathological bone of the ring and elsewhere.

## DISCUSSION

*Cavender (1966)* stated that the caudal peduncles of *Ichthyodectes* and *Gillicus* appeared slightly more constricted than those of *Xiphactinus*, but stated that except where extreme large size indicated a specimen pertained to *Xiphactinus*, the caudal skeletons of *Ichthyodectes*, *Gillicus,* and *Xiphactinus* were indistinguishable. However, *Bardack (1965)* differentiated the vertebrae of *Gillicus* and *Ichthyodectes* from those of *Xiphactinus* on the basis of differences in the dorsoventral height of the lateral ridge between the two

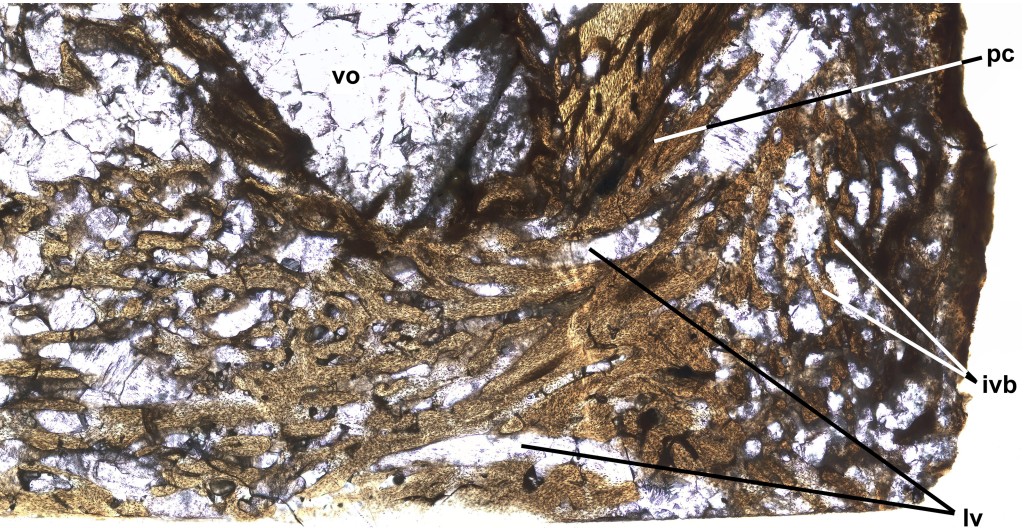

**Figure 6   Intervertebral bone.** Ichthyodectiform fish YPM VP 42619. Detail photograph of the thin section of the centrum of preural vertebra 10 in Fig. 4 showing pathological bone tissue within the centrum and intervertebral disk. Abbreviations: ivb, intervertebral bone tissue; lvc, longitudinal vascular canal; pc, posterior cotyle; and vo, large void. Scale bar = 1 mm.

prominent longitudinal grooves. He noted that in *Xiphactinus* the ridge is twice the width of the grooves, whereas in *Ichthyodectes* and *Gillicus* the lateral ridge is about the width of the grooves. Unfortunately, in YPM VP 42619 it is not possible to assess the relative width of the longitudinal ridge and grooves because of extensive bony overgrowth, so the specimen must be viewed as an indeterminate ichthyodectiform. Comparing the preserved length of the series of 12 vertebrae to an essentially complete mounted 155 cm long specimen of *Ichthyodectes* (USNM 12358; *Bardack, 1965*) suggests that YPM VP 42619 had a fork length of ~1.4 m and, if *Ichthyodectes*, was not quite half grown.

The pathologies of YPM VP 42619 can be categorized as: (1) overgrowing pathological bone tissue that seems to have been deposited on the external surface of the centra; (2) dense cancellous bone in the intervertebral space and within the centrum; and (3) what are interpreted as calcified or ossified soft tissues associated with the haemal and neural canals. The overgrowing pathological bone and the dense cancellous bone of YPM VP 42619 are unlike Tilly bones, *i.e.,* the common hyperostoses of spines, pterygiophores, etc. of fishes, which typically appear as greatly swollen parts of bones. Although Tilly bones exhibit a characteristic swollen appearance, their histology varies considerably: in *Euthynnus* they were described as consisting of greatly thickened lamellar cortical bone surrounding spongy medullary bone (*Béarez, Meunier & Kacem, 2005*); in *Caranx* as acellular bone (*Chanet, 2018*); and in *Prionotus* as spongy bone (*Meunier, Bearez & Francillon-Vieillot, 1999*). Interestingly, in unidentified fossil fishes from a Plio-Pleistocene paleolake in Tanzania Tilly bones consisted of dense bone with multiple thick annual? layers and have a finely ridged external surface with many small pores opening between the ridges (*Schlumberger & Lucké, 1948*), and the multiple bone layers remind one of the palisade-like layers in YPM

VP 42619 whereas the ridged surface with small pores are somewhat similar to the reticulate surface in YPM VP 42619. Despite the histological variation, in most cases, the histology of Tilly bones, other than being greatly thickened, appears similar to that of the adjacent unaffected bone tissue. It has been suggested that Tilly bones are a response to abnormal environmental conditions (*Schlumberger & Lucké, 1948*). Tilly bones of the unidentified fishes from a Plio-Pleistocene paleolake in Tanzania had high levels of fluorine, and it was suggested that the fishes dealt with high levels of fluorine in the lake's waters by increasing the quantity of bone tissue in the body and depositing fluorine within the hyperostotic bone tissue. However, there is sufficient variation in the histology of Tilly bones that it seems likely that they can result from more than one cause. Regardless, the overgrowing bone of YPM VP 42619 does not exhibit the swollen appearance typical of Tilly bones, and the histology of the pathologies, both in their external appearance and in thin section, differs from that of adjacent unaffected bone tissue, whereas in Tilly bones, though swollen are similar to adjacent unaffected bone tissue.

The overgrowing bone of YPM VP 42619 somewhat resembles diffuse idiopathic skeletal hyperostosis (DISH) and spondylosis deformans of mammals, which often affect the vertebral column of older individuals (*Kuperus et al., 2020*; *Kranenburg et al., 2011*), but both are apparently unknown in fish (J Harland, pers. comm., 2024). DISH is characterized by formation of a continuous solid bone bridge ventral to the vertebral centrum and bridging an unaffected intervertebral disc, whereas spondylosis deformans consists of small bone spurs to bone bridges across the disc space but not covering the entire ventral surface of the centrum. The overgrowing bone of YPM VP 42619 differs from DISH and spondylosis deformans in that it is not confined to the ventral surface of the vertebrae, and in humans and dogs, DISH normally occurs in older individuals, whereas the ichthyodectiform fish in this study seems to have been less than half grown. DISH is usually visualized by X-ray, and it is difficult to compare both the three dimensional fossil and the histological thin section of YPM VP 42619 to such images. However, *Kranenburg et al.* (*2011*: figure 5) illustrated a sagittally sectioned series of fused vertebrae of a dog exhibiting DISH and also histological sections of vertebrae exhibiting DISH and spondylosis deformans, and it is apparent that the pathological bone bridging between vertebral centra is cancellous bone very similar to the normal bone tissue within the centra, which is unlike what is seen in YPM VP 42619.

The slender subparallel canals with occasional rounded expansions seen in the palisade-like layers of the overgrowing bone of YPM VP 42619 bring to mind the hyphae of fungi (Fig. 5B), so the possibility that the pathologies of YPM VP 42619 could have resulted from a fungal infection should be considered. Fungal infections of the spine of humans are rare, but typically develop as combined bacterial and fungal infections of the spine in immunocompromised humans with fungus in both the vertebral centra and intervertebral disks (*Williams et al., 1999*; *Tay, Deckey & Hu, 2002*; *Caldera et al., 2016*). Studies of the combined infections are usually visualized by X-ray and MRI, which makes comparisons to the three dimensional fossil and the histological thin section of YPM VP 42619 difficult. However, *Cimerman et al.* (*1999*: figure 1) illustrated fungal hyphae within femoral bone tissue, which appear quite similar to the slender subparallel canals of YPM VP 42619, but do not exhibit any regular or subparallel arrangement. Two species of the fungal

genus *Aspergillus* were cultured from the tissue and exhibited hyphae and vesicles ranging from 20 to 40 μm in diameter. Given the similarity of the fungal hyphae and the slender subparallel canals of YPM VP 42619 and the fact that fungal infections of the centrum and intervertebral disk occur in humans, the pathologies of YPM VP 42619 are tentatively interpreted as resulting from a combined bacterial and fungal infection, in which the presence of different infectious agents may explain the different morphologies of the pathologies.

The 11 preural vertebrae and one ural vertebra of YPM VP 42619 represent ~14% of the fish's fork length, and presumably it was the most important 14% in regard to function of the caudal fin in propulsion because lateral flexion of the caudal peduncle adjusts the angle of the caudal fin as it moves through the water so as to maximize thrust. The fusion of the centra would have had a marked adverse affect on the fish's swimming ability. In addition, the calcification and/or ossification of the tissues associated with the neural and haemal canals may have impaired the blood supply to and innervation of the caudal region leading to paralysis of muscles and loss of sensation. As a result, although swimming must have been possible for the fish to have lived long enough for the pathology to become widespread, it presumably was inefficient and would have affected the fish's ability to feed. Most likely, the onset of the adverse effects of the pathologies was gradual, but as they progressed, the fish's health would have declined and the positive feedback of reduced feeding ability and declining health would have led to the fish's death.

## CONCLUSION

The series of 12 vertebral centra of YPM VP 42619 exhibit extensive overgrowing pathological bone deposited on and surrounding the centra, cancellous bone that is denser than in the intervertebral space and within the centrum, and what are interpreted as calcified or ossified soft tissues associated with the haemal and neural canals. The pathologies are unlike any previously described in fossil or extant fishes, and although they are somewhat similar to the diffuse idiopathic skeletal hyperostosis (DISH) and spondylosis deformans of mammals, there are significant differences as well, and it is most likely that the pathologies resulted from combined bacterial and fungal infections. As the pathologies developed, they would have adversely impacted the fish's swimming and feeding abilities, and presumably eventually led to the fish's death.

### Institutional Abbreviations

**USNM**     United States National Museum, Washington, D.C., USA
**YPM**      Peabody Museum of Natural History, Yale University, New Haven, Connecticut, USA

## ACKNOWLEDGEMENTS

I thank J. H. Ostrom, M. A. Turner, and D. Brinkman of the Peabody Museum of Natural History for access to specimens under their care and assistance while studying them. I thank editor A. Farke and reviewers J. Harland and C. Fielitz for thoughtful comments that improved the manuscript.

### Funding

The author received no funding for this work.

### Competing Interests

The author declares that they have no competing interests.

### Author Contributions

- S Christopher Bennett conceived and designed the experiments, performed the experiments, analyzed the data, prepared figures and/or tables, authored or reviewed drafts of the article, and approved the final draft.

### Data Availability

The specimen is catalogued as YPM VP 42619 in the collection of the Peabody Museum of Natural History (YPM), Yale University, New Haven, Connecticut, USA.

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
