# Peer review of "Pathological caudal skeleton of an ichthyodectiform fish from the Upper Cretaceous Niobrara Formation of western Kansas, USA"

_PeerJ, doi:10.7717/peerj.17353_

## Round 0.1 · original submission · Minor Revisions

Two reviewers have provided feedback on this manuscript, and both (reflecting different areas of expertise) were quite positive overall. Some minor areas for revision are suggested (especially from Reviewer 1, in terms of highlighting the relative lack of DISH in fish and updating the literature on Tilly bones), but my hope is that the edits should be fairly easy to address.

I have two additional suggestions for your consideration:

1) Methods, Line 65-69: What is meant by "standard methodology" -- provide a reference (e.g., Lamm 2013), at a minimum, especially for those who may not be familiar with these techniques. If comprehensive detail is not available, that's OK, but a bit more information would be useful if available. An examples of another PeerJ paper with methods and citations is https://peerj.com/articles/3183/#p-15 (there are others).

2) Completely optional suggestion: Consider adding a figure of non-pathological vertebrae for comparison. This is not required, but would be helpful to visualize the abnormalities here, especially for those who are not deeply familiar with icthyodectiform vertebrae.

Citations:
Lamm E-T. 2013. Preparation and sectioning of specimens. In: Padian K, Lamm E-T, eds. Bone histology of fossil tetrapods: advancing methods, analysis, and interpretation. Los Angeles: University of California Press. 55-160

This is a nice contribution on a rarely reported abnormality in Cretaceous fish!

·

Basic reporting

This fascinating paper presents a set of pathological fossil fish vertebrae in a clear and unambiguous manner, using a clear and logical structure, with a good set of accompanying illustrations.

The language used throughout is concise, and highly technical where appropriate. Although this is adjacent to my field (which is fish bone zooarchaeology of rather more recent material), the references cited appear consistent and an appropriate range of papers has been used.

I would of course have liked to see my own paper on fish bone pathologies cited (Harland J., Van Neer W. (2018). Weird Fish: Defining a role for fish paleopathology. In: Bartosiewicz, L. & Gal, E. (eds), Care or neglect? Evidence of animal disease in archaeology. pp. 256-275. Oxbow Books.). In this I question whether or not we should categorise Tilly bones as pathological, as in several extant species these thickened growths are found on all individuals upon reaching a certain size (ie modern haddock ventral cleithra). I also classify the types of pathologies typically found in North Atlantic bone material, and this may be of use to the authors here.

There was no associated raw data here, but the images are high quality and the annotations and captions are extensive and are helpful to the reader.

The paper is self-contained, following a typical structure through to conclusion.

Experimental design

The paper has a detailed description of the pathological bones, followed by an exploration of their aetiology. This may not by typical of the experimental design published in other papers, but for the purposes of this paper it works well.

The subject of fish bone pathologies is niche and under-studied, whether the bones are archaeological (my specialty) or paleontological. Here, the authors have clearly defined the question and they are certainly filling a knowledge gap. They use a rigorous approach to explaining the pathologies present. My initial thoughts regarding this pathology also turned towards DISH, and I was gratified to see this thoroughly explored here. That said, I've never seen DISH presenting in any fish species in archaeological material or in (the very sparse) literature on modern fish pathologies, and it would strengthen the argument to reflect further on this absence. The parallels with human bone pathologies are well placed, particularly the fungal and bacterial infections, and this makes good use of the literature.

The methods used appear to be appropriate, but here it would be better for a fossil fish expert to comment.

Validity of the findings

The findings in this paper are valid and are the aetiology of the pathologies are thoroughly explored in the text. The photographs are of high quality and are well annotated. The conclusions are robust and logical, from the perspective of a zooarchaeologist. However, it would be good to ensure that a fossil fish expert's review is also provided, as these species are not ones I am familiar with.

Additional comments

I have a fascination with fish bone pathologies from the point of view of North Atlantic zooarchaeology, and so I was delighted to have the opportunity to review this paper. These bones are unlikely anything I've seen before, and the explanation and illustrations are very convincing. The analogy with DISH is again unlikely anything I've seen before but it is convincing, as is the use of literature from human bone infections.

My only suggestions for very minor revisions:
1 to update the text on Tilly bones to reflect their non-pathological status in most - but not all - cases, perhaps referencing Harland and Van Neer 2018
2 to add a comment about the lack of DISH-like pathologies in any modern or archaeological fish species, and reflect on this

·

Basic reporting

No Comment

Experimental design

No comment

Validity of the findings

No comment

Additional comments

The author provides a very detailed description of the fossil fish specimen and the osteological pathologies associated with it. This includes a microscopic histological examination of one of the vertebrae. He then compares the observations with known bone pathologies both in fossil and living vertebrates ruling out possible known causes for the pathology. Based on the histological evidence, he concludes what the most likely cause of the pathology was. It sets the stage for further investigations both with fossil and living fish.

I do not feel that there are any major issues in this study and the manuscript warrants only a few minor revisions. I have made specific comments in the manuscript to be addressed.

My only main comment is the placement of figure citations in the text. The author needs to cite the figures more in the text. In several paragraphs, there is either no figure citation or it is not until deeper in the paragraph. For example, there are no figure citations in the paragraph starting on line 125. They would be useful to the reader because this paragraph first describes the sinuous grooves. The paragraph that begins on line 145 is an example where the figure citation is placed in the second sentence and not the first.
One other note. The figure citations in the text uses numbers (example: Figure 3.1) for parts of the figures whereas the captions and the photographs use letters (example: 3A)

---

## Round 0.2 · accepted · Accept

Thank you for your close attention to the comments from the first version of the manuscript. It is, in my view, ready to proceed to publication.